# Development and Application of Automated Sandwich ELISA for Quantitating Residual dsRNA in mRNA Vaccines

**DOI:** 10.3390/vaccines12080899

**Published:** 2024-08-08

**Authors:** David A. Holland, Jillian Acevedo-Skrip, Joshua Barton, Rachel Thompson, Amy Bowman, Emily A. Dewar, Danielle V. Miller, Kaixi Zhao, Andrew R. Swartz, John W. Loughney

**Affiliations:** 1Analytical Research & Development, Merck & Co., Inc., Rahway, NJ 07065, USA; jill_acevedo@merck.com (J.A.-S.); joshua.barton1@merck.com (J.B.); rachel.thompson2@merck.com (R.T.); amy_bowman@merck.com (A.B.); 2Process Research and Development, Merck & Co., Inc., Rahway, NJ 07065, USA; emily.berckman@merck.com (E.A.D.); danielle.miller2@merck.com (D.V.M.); kaixi.zhao@merck.com (K.Z.); andrew.swartz@merck.com (A.R.S.)

**Keywords:** mRNA, dsRNA, sandwich ELISA, high throughput, automation

## Abstract

The rise of mRNA as a novel vaccination strategy presents new opportunities to confront global disease. Double-stranded RNA (dsRNA) is an impurity byproduct of the in vitro transcription reaction used to manufacture mRNA that may affect the potency and safety of the mRNA vaccine in patients. Careful quantitation of dsRNA during manufacturing is critical to ensure that residual dsRNA is minimized in purified mRNA drug substances. In this work, we describe the development and implementation of a sandwich Enzyme-Linked Immunosorbent Assay (ELISA) to quantitate nanogram quantities of residual dsRNA contaminants in mRNA process intermediates using readily available commercial reagents. This sandwich ELISA developed in this study follows a standard protocol and can be easily adapted to most research laboratory environments. Additionally, a liquid handler coupled with an automated robotics system was utilized to increase assay throughput, improve precision, and reduce the analyst time requirement. The final automated sandwich ELISA was able to measure <10 ng/mL of dsRNA with a specificity for dsRNA over 2000-fold higher than mRNA, a variability of <15%, and a throughput of 72 samples per day.

## 1. Introduction

The first use of mRNA to deliver a target protein in vivo was published in 1990 [1] and the use as a therapeutic in 1992 [2]. In the following decades, mRNA-based therapeutics were implemented for use in genetic engineering [3], regenerative medicine [4], protein replacement [5], cancer immunotherapy [6,7], and vaccines [8,9]. These applications have been reviewed in refs. [10,11]. Perhaps most famously, lipid-encapsulated mRNA was utilized in the recent successful development of SARS-CoV-2 vaccines [12,13]. The use of mRNA vaccines is particularly attractive because of the ability to generate on-demand transcript sequences using platformed manufacturing methods to enable rapid response to emergent needs [14]. The recent increase in mRNA vaccine research and development requires the adoption and implementation of new manufacturing techniques, as well as the proper control mechanisms to monitor them [15].

mRNA vaccines encoding an antigen of interest are synthesized by in vitro transcription (IVT) using a linearized DNA template and an RNA polymerase (RNAP) such as T7, SP6, or T3 [15]. T7 RNAP is most commonly used due to its high mRNA yield and fidelity, but T7 RNAP may also produce multiple IVT reaction byproducts, including immunostimulatory double-stranded RNA (dsRNA) [16]. In contrast to the desired immunostimulatory effect of mRNA, which induces a strong adaptive response via antigen-presenting cell activation [14], dsRNA can inhibit ribosomal protein translation, resulting in a 10- to 1000-fold decrease in target protein production [17]. The dsRNA byproducts can generally be classified as short dsRNA, formed from the random annealing of abortive transcripts, and full-length dsRNA, formed from either loop-back elongation or spurious transcription from the non-coding strand of the DNA template [17].

dsRNA is an innate immune system activator which activates cytosolic receptors such as RIG-I, MDA5, and PKR and endosomal receptor TOR3 [14]. Collectively, the immune system activation creates efficacy concerns due to decreased ribosome expression and toxicity concerns due to cytokine secretion. Different aspects of dsRNA contribute to unique innate immune responses; RIG-I recognizes the 5′PPP end and is thus more sensitive to higher concentrations of short dsRNA, whereas MDA5 involves filament formation driven by longer dsRNA [14]. To our knowledge, no regulatory pharmacopeia has provided target acceptable levels of dsRNA in mRNA vaccines. Existing guidance indicates that dsRNA impurities should be removed to the extent feasible, and any residual dsRNA should be well characterized and controlled [18]. A reduction in dsRNA can be achieved by the optimization of the IVT reaction, such as the use of N1-Me-pseudo UTP, codon optimization, and lowering the Mg^2+^ concentration [19], and implementing purification strategies, such as ion-pairing reversed-phase HPLC (IP-RPHPLC) [17] and cellulose purification [20]. To implement these methods to minimize residual dsRNA in mRNA manufacturing, a sensitive and quantitative dsRNA analytical method is necessary.

The strategies to quantitate dsRNA involve cell-based assays, gel electrophoresis assays, chromatography-based methods, and immunochemistry assays [20,21]. Cell-based assays offer a direct measurement of innate immune activation but are complicated, laborious, and expensive. Gel electrophoresis-based assays can be used for characterizing dsRNA but are only semi-quantitative and lack sensitivity. Quantitation by IP-RPHPLC involves the digestion of mRNA followed by the successive resolution of intact or degraded dsRNA and offers the ability to both characterize and quantitate dsRNA [21]. However, this method is limited by high cost and high level of expertise. The heterogeneity of residual dsRNA and presence of mRNA at orders of magnitude higher concentration than dsRNA precludes some methods of measuring nucleotides such as qPCR and next-generation sequencing. 

A 1991 publication by Schönborn et al. described four antibodies (J2, J5, K1, and K2) that specifically bind dsRNA without binding single-stranded RNA and are highly sensitive for a wide range of sequences that are 40 bp in length [22]. These antibodies have been used in several quantitative immunochemical assays such as the dot blot assay, [20] sandwich ELISA [16,23], and lateral flow immunoassay [24]. The sandwich ELISAs offer an appealing strategy for dsRNA quantitation because they offer an excellent balance of sensitivity, throughput, and simplicity and can be implemented using standard equipment found in most labs. Previous publications have presented data generated using a dsRNA sandwich ELISA but do not provide a detailed description of the development and final protocol for the assay [16,23]. To our knowledge, the full end-to-end development of an automated dsRNA sandwich ELISA has not been described in the literature. The objective of this manuscript is to provide a template for implementing a dsRNA sandwich ELISA to support mRNA vaccine manufacturing process development using commercially available J2 and K2 antibodies along with other readily available materials and reagents. To this end, we describe the development and application of an ELISA for quantifying residual dsRNA in non-adjuvanted mRNA vaccine samples ranging from crude IVT reaction products to fully purified bulk drug substance, as well as “lessons-learned” during assay development. We describe the identification and mitigation of assay interference by common IVT reaction components and the optimization of ELISA reagents, report the assay performance, provide examples of the application of the dsRNA, and finally demonstrate success in automating the ELISA in a high-throughput format (Appendix A).

## 2. Materials and Methods

### 2.1. Materials

Tris (hydroxymethyl)aminomethane, Sodium Chloride (NaCl), and Ethylenediaminetetraacetic acid (EDTA) were purchased from Invitrogen (Waltham, MA, USA). Sodium Hydroxide was purchased from Fisher Chemical™ (Pittsburgh, PA, USA). K2 and J2 dsRNA ELISA antibodies and polyinosinic–polycytidylic acid (poly I:C) positive control were purchased from Exalpha (Shirley, MA, USA), 142 bp dsRNA reference standard from Jena Bioscience (Jena, Thuringia, Germany), sample diluent from Cygnus (Southport, NC, USA), and secondary antibody from Jackson ImmunoResearch Laboratories (West Grove, PA, USA). Assay diluent was purchased from Teknova (Hollister, CA, USA), phosphate buffer saline from Cytiva (Marlborough, MA, USA), Tween 20 from MilliporeSigma (Burlington, MA, USA), and 4-methylumbelliferyl phosphate (4-MUP) from Virolabs (Chantilly, VA, USA). Black 96-well MaxiSorp^®^ ELISA plates were purchased from Nunc (Roskilde, Denmark).

### 2.2. Quantitation of dsRNA

To quantitate residual dsRNA in mRNA samples, an ELISA was developed utilizing a Tecan EVO liquid handler configured with a Tecan Infinite F500 plate reader. Unless otherwise noted, assay diluent (Teknova P1391; 10 mM phosphate, 154 mM NaCl, 1% BSA) was used throughout. All incubation steps listed below were performed with plates left uncovered, without shaking at room temperature. Washing was performed with 3 wash cycles of 400 µL wash buffer (10 mM phosphate 154 mM NaCl, 0.05% Tween-20 pH 7.0) using either a Tecan HydroSpeed or BioTek EL406 plate washer. Unless otherwise stated, 96-well Black Maxisorp plates (Nunc 437111) were coated with 2 µg/mL mouse anti-dsRNA J2 IgG (Exalpha 10010500) in coating buffer (Hyclone SH29882925; 10 mM phosphate, 154 mM NaCl pH 7.0) per well for 1 h. Coating antibody was then aspirated prior to blocking with assay diluent for at least 15 min and no longer than 8 h. Blocked plates were washed as described above. Unless noted otherwise, mRNA samples were pre-diluted 2-fold in Sample Dilution Buffer (Cygnus I028) prior to serial diluting in Sample Dilution Buffer. After a 2 h sample incubation, plates were washed and incubated with mouse anti-dsRNA K2 IgM (Exalpha 10030010) for 1 h; unless otherwise stated, a 15-fold dilution of K2 IgM was used throughout. Plates were washed and then incubated with alkaline phosphatase (AP)-conjugated goat anti-mouse IgM (Jackson ImmunoResearch 115-056-020) for 1 h; unless otherwise stated, 0.3 µg/mL of conjugate antibody was used throughout. Samples were resolved using 4-MUP (Virolabs XPHOS-100) as the substrate, and the signal was detected by fluorescence at 360 nm excitation and 465 nm emission using a Tecan Infinite F500. The addition of capture antibody, blocking reagent, detection antibody, AP-conjugated secondary antibody, and 4-MUP substrate was performed by a single-step transfer utilizing the 96-well multichannel arm on the Tecan EVO. Eight 2-fold serial dilutions of mRNA samples were performed using a single column of tips on the Tecan EVO 96-well multichannel arm. Determination of dsRNA concentration in mRNA samples was calculated using parameters determined from a four-parameter logistic (4PL) fit of a standard curve generated with 142 bp dsRNA purchased from Jena Bioscience (RNT-sci-10080100). To correct for differences in total mRNA concentration of in-process samples, the concentration of dsRNA is normalized to the total mRNA concentration and reported as the percentage of dsRNA (*w*/*w*). 

### 2.3. Preparation and Purification of mRNA Containing Samples

mRNA vaccine constructs were transcribed from linear DNA templates with encoded poly(A) tails following a standard IVT reaction protocol that uses a combination of nucleotide triphosphates (NTPs), T7 RNAP, pyrophosphatase, RNase inhibitor, and a Tris buffer system containing DTT, MgCl_2_, and CleanCap AG, as previously described [25]. The IVT reaction was incubated for 3–4 h at 37 °C followed by enzymatic digestion of the DNA template with DNase I at 37 °C for an additional 30–60 min. An optional enzyme removal step was included in selected samples by proteinase K/SDS addition after enzymatic digestion of the DNA template step. After the IVT reaction and digest steps, enzyme activity was quenched by the addition of 50 mM EDTA. 

Samples that were not purified are referred to as IVT samples. Selected samples were purified using an mRNA purification process involving oligo dT chromatography, tangential flow filtration for concentration and buffer exchange into an mRNA storage buffer, and bioburden reduction filtration [25] to generate bulk drug substance (BDS) samples. IVT samples, purification process intermediate samples, and BDS samples were analyzed by the dsRNA assay. Two samples were subjected to additional cellulose purification (Section 3.4).

The total concentration of RNA was determined using a Nanodrop 8000 Spectrophotometer (ThermoScientific, Waltham, MA, USA) using ThermoScientific software V2.3.3. The instrument was blanked with subsequent measurements of 2 µL RNase-free water followed by 2 µL sample-specific matrix buffer. After blanking, the absorbance at 260 nm of the 2 µL sample was measured and converted into RNA concentration using an OD equivalence of 40.

The impact of selected IVT reaction and purification process parameters on mRNA yield and purity was investigated. For example, a concentrated stock of T7 RNAP was used in one IVT reaction condition resulting in additional matrix carry over relative to the standard usage practice. The additional matrix carried over into the IVT reaction is differentiated from the standard IVT reaction conditions below by referring to the former as “IVT Buffer 2” and the latter as “IVT Buffer 1”. As another iteration of the IVT reaction, several additives (numbered 1 to 3) were also included in the IVT reaction to evaluate the robustness of the IVT reaction and effect on dsRNA production by the T7 RNA polymerase when these additives were included. Several iterations were collectively incorporated into a procedure referred to below as “procedure 2” to distinguish against an earlier utilized “procedure 1”. These process changes may include a change in manufacturing facility, change in reagent sources, and an optimization of purification parameters within the general mRNA purification strategy outlined above.

## 3. Results

### 3.1. Assay Interference by IVT Reagents

As a preliminary proof of concept, the newly developed dsRNA ELISA method was utilized to measure residual dsRNA within in-process vaccine samples which included both IVT reaction samples and purified mRNA samples. These samples were measured as described in Section 2 with the following exceptions: detection was performed with a 2-fold dilution of mouse anti-dsRNA K2 IgM (K2 Ab) and serial dilutions were performed in Tris-buffered saline with 1 mM EDTA (STE Buffer: 25 mM Tris, 130 mM NaCl, 27 mM KCl, 1 mM EDTA pH 7.4) and were performed manually using an eight-channel electronic pipette.

We report first the purified samples as an example of a well-behaved sample. The dose–response curves of both purified samples have a sigmoidal curve that is visually parallel to the reference standard (Figure 1A). The apparent concentration of residual dsRNA in the purified samples was back-calculated from the fluorescence obtained from a four-parameter logistic (4PL) fit of the 142 bp dsRNA reference standard (Figure 1B). Contrary to the elution samples, the analysis of the IVT samples resulted in dose–response curves with a prominent hook at high dsRNA concentrations (Figure 1C) and a high level of uncertainty in the apparent concentration (CV > 70%; Figure 1D). These results suggest a ~5-fold clearance of dsRNA in purified samples; however, the non-parallel curves leave a high level of uncertainty in the concentration of dsRNA in the unpurified IVT samples. We hypothesized that the prominent hooking observed with IVT samples was due to assay interference from one or more components in the residual IVT reaction mixture. 

To interrogate the potential of assay interference, a “mock IVT sample” was prepared by spiking 142 bp dsRNA into the IVT reaction mixture to a target dsRNA concentration of 1000 ng/mL. The mock IVT sample was subjected to quantitation by dsRNA ELISA while performing serial dilution either in STE buffer (Figure 2A: red) or in STE buffer containing IVT mixture (Figure 2A: blue). See Appendix A for schematic. The ELISA in which the serial dilution was performed in STE buffer with the IVT mixture resulted in no signal over the entire dose–response curve, indicating a significant level of interference by the IVT mixture (Figure 2A: blue). The ELISA in which the serial dilution was performed in STE buffer resulted in a dose–response curve characterized by a steady increase and then subsequent decrease (Figure 2A: red). The dose–response curve could be partially recovered by allowing the IVT mixture to be diluted out; however, the effect of the IVT mixture significantly perturbs the overall curve shape. Collectively, these two results indicate that one or more components of the IVT mixture were interfering with the dsRNA ELISA.

To verify that the results in Figure 2A were not an artifact of the experimental design, a complementary study was performed in which a “mock purified sample” was prepared by spiking 142 bp dsRNA into a final bulk drug substance (BDS) buffer to a target dsRNA concentration of 1000 ng/mL. The dsRNA ELISA was performed on the mock purified sample either by serially diluting it in STE buffer (Figure 2B; red) or in STE buffer with BDS buffer (Figure 2B; blue). Contrary to the mock IVT sample, the mock purified sample produced a sigmoidal dose–response curve irrespective of the presence of a BDS matrix. These results indicate that the dsRNA ELISA is compatible with the BDS matrix. The observation of matrix interference in the mock samples prepared with IVT matrix and not in the mock samples prepared with BDS buffer indicates that the assay interference is derived from a component that is unique to the IVT matrix.

To understand the cause of the interference, we evaluated the IVT reaction mixture and identified the commonly added enzyme removal components, proteinase K and SDS, as the probable interfering components. Proteinase K and SDS can interfere with the ELISA by degrading the coating antibody by proteolysis or denaturation, respectively. The IVT sample prepared without the enzyme removal step did not demonstrate the same interference patterns (Figure 2C) as the IVT samples with the enzyme removal step (Figure 1B). All subsequent IVT samples evaluated using the dsRNA ELISA were prepared without the enzyme removal step.

### 3.2. Optimization of ELISA Antibodies

With the successful proof of concept for quantitating residual dsRNA in IVT and purified mRNA samples, the dsRNA ELISA was further optimized to improve assay performance and reduce cost. The dsRNA ELISA of a 1000 ng/mL 142 bp dsRNA reference standard in STE buffer was performed using plates that were coated with either 2 (Appendix A), 4 (Appendix A), or 8 µg/mL (Appendix A) of mouse anti-dsRNA J2 IgG; using either PBS (panels A–C) or NH_4_SO_4_ (panels D–F) as the coating buffer; and with the K2 detection antibody diluted by 2-fold (black), 4-fold (red), 8-fold (blue), or 16-fold (green). Each condition was measured in duplicate, and the resulting dose–response curves were averaged together before fitting to a 4PL curve. No trends were observed in terms of the slope, EC-50, background, or signal-to-baseline ratio (S/B) as a function of K2 detection antibody or between the use of PBS or NH_4_SO_4_ coating buffer (Figure 3). The signal-to-baseline ratio trended towards decreasing values with increasing J2 coating antibody concentrations in concordance with an increase in blank value. The lowest EC-50 value (tightest antigen binding) was observed when 2 µg/mL of J2 antibody was coated in PBS buffer.

The final conditions selected were 2 µg/mL J2 (coating Ab) in PBS and a 15-fold dilution of K2 as the detection antibody. Due to having the lowest EC-50 value, lowest blank value, and highest signal-to-baseline ratio, 2 µg/mL J2 (coating Ab) in PBS was selected. The lowest concentration of K2 Ab was selected to reduce the overall cost per assay.

### 3.3. Performance of the dsRNA ELISA

A partially purified mRNA was selected for use as a positive control. The apparent concentration of dsRNA in the positive control was determined to be 1347 ± 96 ng/mL (average and one standard deviation) from six replicate measurements performed on a single plate. The variability (7% CV) is taken to represent the intra-plate repeatability. A similar apparent concentration (1288 ± 136 ng/mL) was determined from eighteen replicate measurements across three plates, although the variability was slightly higher (11% CV). The 11% CV value is taken to represent the inter-plate repeatability of the dsRNA ELISA. A total of 81 replicate measurements of the positive control, performed on 14 separate days, across two analysts, determined that the concentration of dsRNA was 1429 ± 276 ng/mL. The higher CV value observed for this final assessment (21% CV) is attributed to the day-to-day variability in the assay. 

The accuracy was assessed by the measuring spike recovery of 0, 400, 800, 2000, 5000, or 10,000 ng/mL 142 bp dsRNA reference standard (RS) in a fixed background of 1 mg/mL mRNA. The 1 mg/mL mRNA used for the background contained 390 ng/mg residual dsRNA before the dsRNA spike. The apparent concentration was within 22% of the expected value after accounting for the contribution of residual dsRNA in the 1 mg/mL mRNA (Table 1). The data were fitted to a linear equation with a slope of 1.09, a y intercept of 552, and a coefficient of determination (R2) of 0.997. These data demonstrate that the assay is accurate for quantitating dsRNA in a range of ~400–10,000 ng/mL in the presence of excess mRNA. The general linearity observed suggests little to no interference of the dsRNA ELISA by mRNA.

Assay selectivity was assessed based on the accuracy of dsRNA quantitation in a background of mRNA ranging from a 250- to 1000-fold excess of the dsRNA concentration. To this objective, a fixed concentration of 2000 ng/mL 142 bp dsRNA RS was spiked into an mRNA sample where the concentration was 0.5, 0.75, 1.0, 1.5, or 2.0 mg/mL mRNA (Table 2). The concentration of dsRNA contributed from the mRNA was determined by measuring the undiluted mRNA used to prepare the samples. After correcting for the residual dsRNA contributed from the mRNA sample, the apparent concentration of dsRNA of each prepared mRNA sample was 109 to 124% of the expected value. The deviation from the expected value is attributed to a cumulative effect of pipetting error and assay variability. The lack of a systematic divergence from expected values as a function of mRNA concentration suggests that the mRNA is neither interfering with nor cross-reacting with the dsRNA ELISA.

### 3.4. Application of the dsRNA ELISA

The dsRNA concentration alone is insufficient to inform best manufacturing practices because the mRNA concentration of in-process samples can vary in mRNA concentration from 0.1 mg/mL to ≥4 mg/mL depending on the processing stage as well as variability in sample preparation. In order to best compare the concentration of dsRNA from one sample to another, the concentration of dsRNA is normalized to the total RNA concentration and reported as the relative percentage of dsRNA. To evaluate the ability of the dsRNA ELISA to differentiate between mRNA samples with a high or low relative percentage [dsRNA] in a real-world application, an IVT sample was generated using unmodified UTP instead of N1-Me-psuedo UTP. N1-Me-psuedo UTP has been previously shown as a substitution for unmodified UTP which significantly reduced residual dsRNA in IVT products [19]. The dsRNA ELISA reported 20-times higher residual dsRNA in the IVT mRNA sample produced with unmodified UTP compared to the IVT mRNA sample produced with N1-Me-pseudo UTP (Figure 4A). The successful trending of high and low residual dsRNA containing mRNA samples indicates that the dsRNA ELISA performed in this study is sufficient for head-to-head comparisons of mRNA samples and for the general trending of results. 

The dsRNA ELISA was used to compare the amount of residual dsRNA in seven mRNA vaccine constructs purified using two different processes, 1 and 2 (Figure 4B). The concentration of residual dsRNA was between 0.015 and 0.11% dsRNA for all samples tested (Figure 4B). Overall, the residual dsRNA measured in mRNA samples prepared with “process 2” was equal to or lower than that in “process 1” indicating a general improvement in mRNA production following process 2 (Figure 4B). 

The average concentration of dsRNA in constructs 1–5 was the same (within assay error) as constructs 6 and 7 despite a ~2-fold difference in mRNA length, suggesting that the dsRNA ELISA result is not dependent on the mRNA sequence length (Figure 4B). A potential trend in the sequence-specific concentrations of residual dsRNA is observed as constructs 2–6 seem to follow a similar trend regardless of whether the mRNA was manufactured using process 1 or process 2. Of notable exception is construct 1, which differs significantly between the two processes, with the mRNA prepared by process 1 containing significantly more dsRNA than the comparable mRNA prepared by process 2. The drug substance residual dsRNA content for some sequences may be more sensitive to manufacturing process changes. 

The dsRNA ELISA was used to support the upstream and downstream development of the mRNA manufacturing process. Residual dsRNA is a critical component in monitoring mRNA manufacturing, and thus, quantitation is critical for ensuring the proper optimization of manufacturing practices. The dsRNA ELISA method was evaluated across multiple representative process development applications (Figure 5). 

The dsRNA ELISA was used to evaluate the benefit of a buffer change during the IVT reaction (Figure 5A). The IVT process was evaluated using two different IVT reaction buffers. IVT Buffer 2 had a positive increase in the overall mRNA yield. However, the dsRNA ELISA showed that using Buffer 2 for the IVT reaction also increased the relative percentage of dsRNA by 6-fold. The relative percentage of dsRNA is an important factor to consider during process optimization and has been used to evaluate additional IVT reaction mixtures (Appendix A). 

Given the immense range of conditions that can be optimized for mRNA IVT, automating the IVT reaction offers a tremendous advantage over manual processing. The dsRNA ELISA was utilized to evaluate whether automated processing (solid) produced comparable results to manual processing (striped) in terms of residual dsRNA production (Figure 5B). No significant difference in residual dsRNA was observed for IVT mRNA produced using manual processing or automation after either 1.5 h or 3 h of reaction time. This demonstrates that automation does not increase the dsRNA produced in the IVT reaction. The targeted removal of residual dsRNA performed by two cellulose-based purification strategies listed as “Purified 1” and “Purified 2” is demonstrated using an mRNA sample intentionally manufactured to have a high relative percentage of dsRNA (Figure 5C). Both strategies were shown to be individually capable of removing ≥95% of the residual dsRNA from a common load sample. 

The dsRNA ELISA was used to detect residual dsRNA across three steps of a standard mRNA purification protocol including oligo dT chromatography, tangential flow filtration, and membrane filtration (Figure 5D). In each of the five constructs tested, the amount of dsRNA present prior to purification is less than 0.05% of the total mRNA. No reduction in the residual dsRNA is observed across the three purification steps for the five constructs (Figure 5D), demonstrating that these process steps do not separate dsRNA from the mRNA product. Based on the purification methods, the small amounts of residual dsRNA produced in the IVT samples likely are associated with large RNA species containing a poly-A tail.

### 3.5. Automation and Future Development

The results presented thus far were generated using a semi-automated ELISA protocol without end-to-end timing control. As described in Section 2, the semi-automated ELISA utilizes a robotic 96-well multichannel arm (MCA) to perform reagent transfers and serial dilutions but lacks end-to-end timing and uses a discrete plate washer (BioTek) and plate reader (Tecan). An analyst is required to change out reagents, transfer them at the appropriate incubation times, and move the plate to a washer and a reader. By fully automating the dsRNA ELISA, we expect that the time burden on the analyst and the overall assay variability will decrease. Initial efforts at fully automating the dsRNA ELISA (Figure 6A: dashed lines) were met with unsatisfactory results compared to the semi-automated results (Figure 6A: solid lines) performed using identical assay conditions. The fully automated ELISA produced a dsRNA dose–response curve for 142 bp dsRNA RS (Figure 6A; black) and positive control (Figure 6A; red) that was significantly right-shifted relative to the semi-automated ELISA and failed to form an upper plateau (Figure 6A). 

Transitioning the dsRNA ELISA to the fully automated protocol required additional optimization of the ELISA reagent concentrations, as outlined in Appendix A. When the fully automated ELISA was performed using the optimized concentration of the J2 coating antibody, the resulting dose–response curve of 142 bp dsRNA RS (black) and positive control (red) agreed well with the semi-automated ELISA in terms of the overall curve shape and the EC-50 value of the reference standard (Figure 6B). These data suggest that increasing the coating antibody was sufficient to implement the fully automated ELISA. Automating the dsRNA ELISA reduced the inter-plate repeatability from 11% to 5% and the day-to-day variability from 21% to 13%.

## 4. Discussion

Commercially available dsRNA ELISA kits are available which could potentially support dsRNA quantitation at a small scale; however, a sandwich ELISA using bulk reagents is required to handle sample testing at a large scale. The automated ELISA described in this paper can support the testing of 72 samples per day (with a full titration curve/sample) with only a half-day analyst obligation. Within the past year, an in-house dsRNA sandwich ELISA has been utilized to analyze residual dsRNA in over 500 mRNA samples. Supporting 72 samples per day with a full titration curve using a typical five-plate commercial kit would require opening and combining three kits which would impart additional variability and analyst burden from preparing reagents from multiple sources. Furthermore, the cost per sample of testing with a kit is over twice as much as testing with the in-house dsRNA ELISA described in this paper. Two commercially available ELISA kits were evaluated in our lab and were seen to have unsatisfactorily high variability (Appendix A). This variability was suspected to be due to an inadequate amount of coating antibody included with the kit. More pertinently, the format of a kit precludes reagent optimization, which was found to be critical in the development of the dsRNA ELISA using bulk reagents. Furthermore, using a kit constrains the number of samples which can be tested within a study, as using multiple kits could contribute to variability. When compared to commercially available dsRNA ELISA kits, an in-house sandwich ELISA can assay a greater volume of samples, with reduced cost, reduced analyst time obligation, and reduced material waste. 

Seven unique mRNA constructs have been assayed by the dsRNA sandwich ELISA with nucleotide lengths that differ by up to 2-fold (Appendix A). Among these constructs, some are coding sequence variants of the same backbone, whereas others have different coding sequences and untranslated regions. The dsRNA sandwich ELISA has been successfully implemented toward the quantitation of residual dsRNA in partially purified mRNA samples across a range of manufacturing steps including oligo dT chromatography, tangential flow filtration, membrane filtration, and cellulose chromatography. mRNA samples across these manufacturing steps contain a range of background buffer components, including citrate buffer, Tris buffer, 0–600 mM NaCl, and pH values from 6.4 to 7.2. Furthermore, the dsRNA sandwich ELISA has been implemented towards testing unpurified IVT reaction products in a complicated matrix background with a wide range of components including various concentrations of NTPs, DNA template, RNA polymerase, polymerase inhibitor, MgCl_2_, EDTA, DTT, and assorted IVT additives. In each of these conditions, the concentration of residual dsRNA was determined without significant interference from the IVT reaction components. The only IVT reaction components that were observed to completely inhibit meaningful quantitation were the enzyme removal components, proteinase K and SDS (Figure 1B). The wide range of mRNA constructs and matrix backgrounds in the samples tested speaks to the broad applicability and robustness of the dsRNA sandwich ELISA. 

Automating the dsRNA ELISA reduces the time burden of the assay, improves the assay reproducibility, and improves the throughput of continuous assay support. However, establishing an automated ELISA requires a high initial investment in time and expertise. A semi-automated or manual ELISA can provide a lower barrier of entry for use in the quantitation of residual dsRNA in mRNA samples. In this study, a semi-automated ELISA was implemented using resources and protocols readily available at the time, while a fully automated ELISA was being developed. Coating plates with 2 µg/mL of J2 antibody was sufficient for the semi-automated ELISA, whereas the fully automated ELISA required 4 µg/mL (Appendix A). This difference between the semi-automated and fully automated ELISA was partially attributed to the change from the BioTek plate washer used in the semi-automated ELISA to the Tecan HydroSpeed plate washer used for the fully automated ELISA. The BioTek plate washer was equipped with a manifold that contained side washing pins. This manifold was selected based on a historic use of the equipment for washing cell-coated plates and was not anticipated to be a critical feature of the assay. It is thought that the downward washing manifold equipped on the HydroSpeed is harsher and partially removes J2 antibody by sheer force, whereas the side washing allows for a gentler wash that does not disrupt the J2 coating antibody. By oversaturating the plates with J2 coating antibody, the partial removal of J2 antibody from washing does not reduce the remaining J2 below the assay saturation point. The plate washer, manifold, and wash settings are likely to be variable from lab to lab. The washer should be a major focus of attention for new labs adopting the ELISA, likely requiring optimization for each individual system. 

During assay optimization, it was discovered that the mouse anti-dsRNA K2 monoclonal IgM detection antibody could be diluted 15-fold without a noticeable depreciation in assay quality. A large dilution of K2 was logistically advantageous as the K2 IgM monoclonal antibody is provided as a hybridoma supernatant of unspecified concentration, shipped in 10 mL quantities. A typical assay could utilize up to 140 mL in a single 12-plate run. However, it should be noted that the optimization and utilization of K2 detection antibody is based on a single lot of K2 antibody. Variations among lots and suppliers were not evaluated as a part of this study. 

One objective of this publication is to provide a template which could be easily adopted by other laboratories. If equipment availability and/or material cost preclude the adoption of the fluorescence-based reagents used in this study, the assay could be performed with horseradish peroxidase-conjugated goat anti-mouse IgM (Jackson Immunoresearch 115-036-075) as the conjugate antibody and resolved with 3,3′,5,5′-Tetramethylbenzidine (TMB) substrate (Sigma-Aldrich T0440). A commercially available polyinosinic–polycytidylic acid (poly I:C) reagent was evaluated as a potential alternative for either the reference standard or positive control for the dsRNA ELISA. Poly I:C is structurally similar to double-stranded RNA and is commonly used as a synthetic analog to dsRNA in immune research. It was observed that the dose–response curve generated by the poly I:C lacked parallelism with the intended in-house mRNA samples, resulting in a very high level of uncertainty in dsRNA concentration (Appendix A). For this reason, the poly I:C reagent is not recommended for use with the dsRNA ELISA when performed as described in this study. 

The performance of the developed semi-automated ELISA was analyzed in terms of repeatability, precision, sensitivity, selectivity, and accuracy. Overall, the performance was satisfactory for its intended application. Increasing replicates improves precision at the cost of throughput. This decision should be made at a phase-appropriate level and with understanding of the precision required for informed decision making. The spike recovery of the 142 bp dsRNA reference standard indicated that the assay was linear from 400 ng/mL to 10,000 ng/mL in the presence of 1 mg/mL mRNA which equates to the detection of 40–1000 ng dsRNA per 100 µg mRNA. The lower limit of quantitation determined by this approach was constrained by the mRNA samples used to provide the 1 mg/mL background; the true limit of quantitation is likely far lower than presented. Based on a simple dilution of 142 bp dsRNA reference standard, we see that the ELISA can detect as low as 1 ng dsRNA suggesting a very high level of sensitivity. The dsRNA sandwich ELISA was successfully utilized to measure residual dsRNA ranging from 1% to 0.01% indicating a high level of both selectivity and sensitivity. 

Two approaches were utilized to approximate the accuracy of the dsRNA ELISA described in this study: spike recovery of 142 bp dsRNA RS and differentiation of the apparent residual dsRNA concentration in a “high” and “low” dsRNA sample. The spike recovery was measured to be accurate within 30%; the deviation from the expected value is thought to be from variability in pipetting the low volumes implemented for this study. Without assurance that the affinities of the J2 and K2 antibody for dsRNA are fully sequence- and length-independent, the spike recovery of the 142 bp dsRNA RS can only approximate accuracy. Similar to the antibodies used in whole-cell protein assays [26], the J2 and K2 antibodies do not target specific species but instead measure the aggregate of all dsRNA present. The 142 bp dsRNA is a uniform, well-defined sequence, whereas the dsRNA in mRNA samples is likely a heterogenous mixture of various lengths and uncertain sequence. An uncharacterized bias of J2 and K2 binding to dsRNA of a particular sequence or length could skew the apparent concentration as determined by the sandwich ELISA. Furthermore, it is known that the J2 antibody only recognizes sequences greater than 40 bp [22]; the values presented in this paper could be an underestimate of the total amount of dsRNA if the predominant species are short fragments of <40 bp. The detection of a 20-fold increase in residual dsRNA in a mRNA sample produced using unmodified UTP relative to an mRNA sample produced using N1-Me-pseudo UTP gives credence to the accuracy of trending performed using the dsRNA ELISA. This is in agreement with the literature reports of a significant reduction in dsRNA using a N1-Me-pseudo UTP substitution for unmodified UTP [19].

To better demonstrate the accuracy of the dsRNA ELISA when quantitating residual dsRNA in mRNA samples, it is imperative that the assay be investigated for potential affinity differences against mRNA of different sequences and lengths. The use of the uniform 142 bp dsRNA is only suitable for accurately quantitating dsRNA of unknown composition if the assay is shown to be reasonably independent of sequence and length. Otherwise, a bespoke reference standard, informed by the careful characterization of the residual dsRNA in an mRNA product of interest, is required for each individual project. Furthermore, the confirmation of residual dsRNA by an orthogonal method, such as ultraviolet (UV) absorbance spectroscopy following isolation by reversed-phased HPLC [21], would improve confidence in the accuracy of the dsRNA ELISA. A representative dsRNA RS could be produced for each sequence by synthesizing and annealing the complementary RNA strand. However, this increases the complexity to qualify a new RS for each new sequence evaluated.

## 5. Conclusions

As more mRNA-based products come to the market, the regulatory environment around mRNA will become more established, and the ability to accurately quantitate residual dsRNA will be more critical [18]. A sandwich ELISA is known to be highly amendable to good manufacturing practices (GMPs) and is applicable in a quality control environment. Furthermore, a sandwich ELISA is highly modular and is, therefore, well suited to the changing regulatory landscape. For example, a universal reference standard could easily be substituted for the 142 bp dsRNA reference standard, and alternate antibodies could be used in place of the J2 and K2 antibodies. The study presented here describes the development and application of a high-throughput automated dsRNA ELISA for quantifying residual dsRNA in mRNA samples. The dsRNA sandwich ELISA protocol presented here is suitable for use in ensuring product consistency during manufacturing and could serve as a template to be iterated on as the mRNA community transitions to a universal assay to support the flourishing field of mRNA-based therapeutics.

## Figures and Tables

**Figure 1 vaccines-12-00899-f001:**
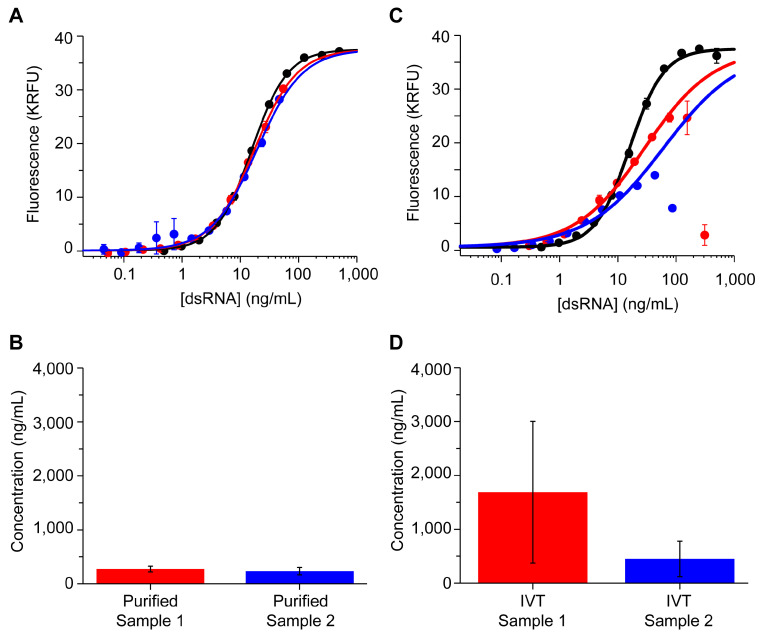
Proof-of-concept assay quantitating residual dsRNA in mRNA elution samples (panel (**A**,**B**)) and IVT samples (panel (**C**,**D**)). Top panels (**A**,**C**) display the dose–response curves obtained for reference standard (black), sample 1 (red), and sample 2 (blue). Bottom panels (**B**,**D**) display the apparent concentration of dsRNA in sample 1 (red) and sample 2 (blue) which were back-calculated from the fluorescence values using the parameters obtained from a 4-parameter logistic fit of the standard curve. Error bars represent 1 standard deviation of two sample replicates.

**Figure 2 vaccines-12-00899-f002:**
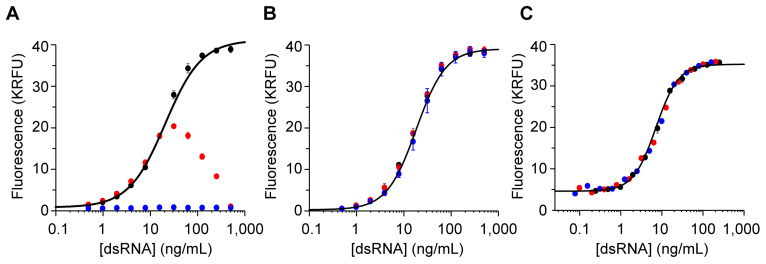
Evaluation of assay interference by components within the IVT reaction mixture (**A**) or the BDS buffer (**B**). The 142 bp dsRNA reference standard was spiked into either IVT mixture (**A**) or formulation buffer (**B**) to prepare a 1000 ng/mL mock IVT or mock BDS sample, respectively. Serial dilutions were performed either in STE buffer alone (red curve) or in STE containing the respective sample buffer (blue). A representative curve measured from 2 IVT samples prepared without the optional enzyme removal step (red and blue) relative to reference standard (black) is demonstrated in panel (**C**).

**Figure 3 vaccines-12-00899-f003:**
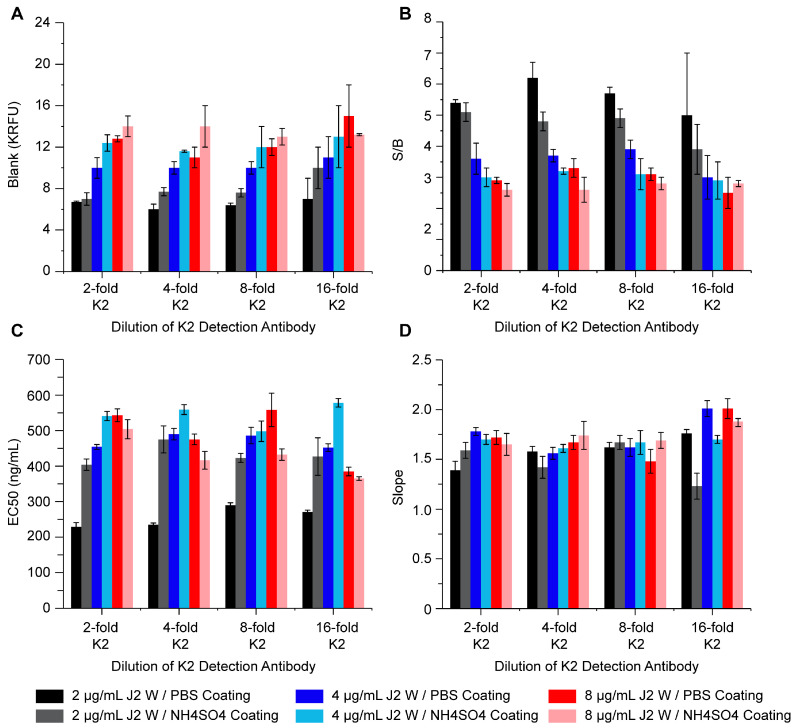
dsRNA ELISA curve parameters obtained from reagent optimization. The blank value (**A**), signal-to-baseline ratio (**B**), EC-50 value (**C**), and the slope of the linear region (**D**) are given for assays performed using either 2 µg/mL, 4 µg/mL, or 8 µg/mL of J2 coating antibody (black, blue, and red, respectively) in either PBS coating buffer (dark color) or NH_4_SO_4_ coating buffer (light color) and with detection antibody diluted 2-fold, 4-fold, 8-fold, or 16-fold. Values reflect the average of 2 replicates; error bars indicate 1 standard deviation.

**Figure 4 vaccines-12-00899-f004:**
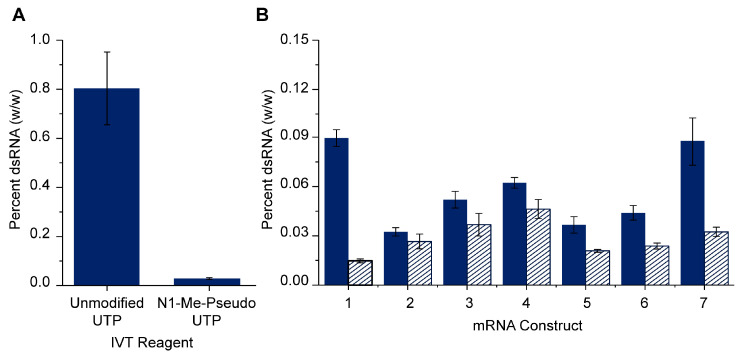
(**A**) Proof-of-concept assay measuring dsRNA concentration in mRNA IVT reaction products performed using either unmodified UTP or N1-Me-pseudo UTP. Values indicate the concentration of dsRNA expressed as the percentage of dsRNA relative to the concentration of mRNA (see Section 2). Average and standard deviation is of 3–4 dilutional replicates. (**B**) Residual dsRNA measured by dsRNA ELISA in mRNA construct 1–7 BDS samples. mRNA BDS samples were prepared using either process 1 (solid bars) or process 2 (striped bars). Values indicate the concentration of dsRNA expressed as the percentage of dsRNA relative to the concentration of mRNA (see Section 2). Average and standard deviation is of 3–4 dilutional replicates.

**Figure 5 vaccines-12-00899-f005:**
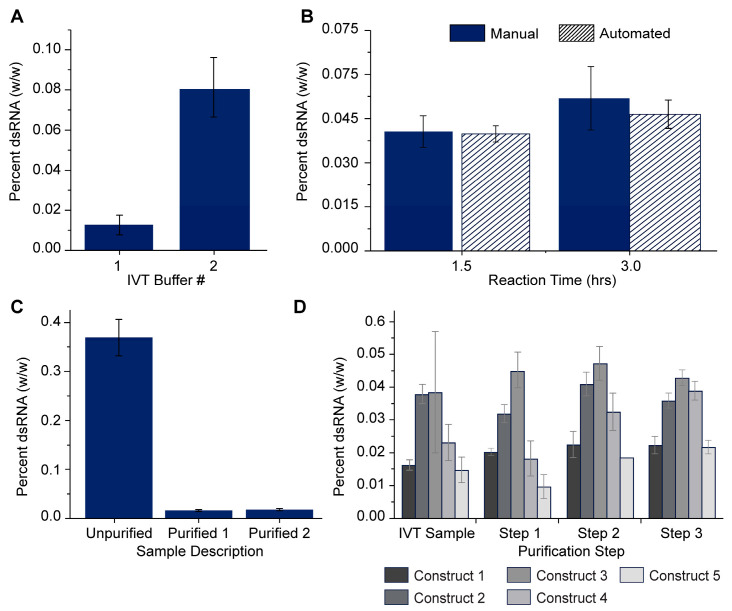
Use of dsRNA ELISA to support process development. (**A**) Evaluation of the effect of two different polymerase buffer conditions on the relative % dsRNA. (**B**) Effect of manual (solid) vs. liquid handler (striped) pipetting on the relative percentage of dsRNA in IVT reaction products after 1.5 h or 3 h reaction time (**C**) Removal of dsRNA using two cellulose-based purification strategies for capturing dsRNA. (**D**) Relative % dsRNA in an IVT sample before and after a commonly used 3-step purification (steps 1–3) including OdT elution, tangential flow filtration, and buffer exchange as measured in constructs 1–5 (grey scale).

**Figure 6 vaccines-12-00899-f006:**
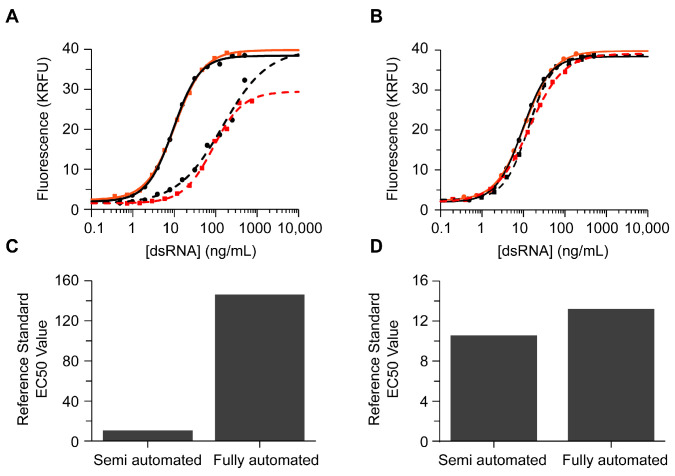
Comparison of ELISA dose–response curve of 142 bp dsRNA reference standard (black circles) and positive control (red squares) generated using the semi-automated ELISA protocol (solid lines) or the fully automated ELISA protocol (dashed lines) before (**A**) and after (**B**) optimization of the fully automated ELISA, as described in Section 3. Lines indicate the fit of raw data to a 4PL curve. (**C**,**D**) EC-50 value measured for the reference standard when using either the semi-automated or fully automated ELISA before optimization (**C**) and after optimization (**D**).

**Table 1 vaccines-12-00899-t001:** Accuracy assessment performed via spike recovery of 142 bp dsRNA reference standard in a 1 mg/mL mRNA background. Expected concentration of dsRNA is reported as the sum of the measured concentration in the 1 mg/mL mRNA and the concentration of spiked 142 bp dsRNA. The expected dsRNA concentration in the 0 ng/mL sample is non-applicable (N/A) because this sample was included as a control to account for the residual dsRNA in the mRNA material.

[142 bp dsRNA Spike] (ng/mL)	Expected [dsRNA] (ng/mL)	Measured [dsRNA] (ng/mL)	% Expected
10,000	10,390	11,000	106%
5000	5390	6400	119%
2000	2390	2800	117%
800	1190	1450	122%
400	790	860	109%
0	N/A	390	-

**Table 2 vaccines-12-00899-t002:** Accuracy and selectivity as assessed by the accuracy of quantifying 2000 ng/mL dsRNA in a background of 0.5–2.0 mg/mL mRNA. Expected concentration of dsRNA is reported as the sum of the dsRNA extrapolated from that measured in the 2.2 mg/mL mRNA used to prepare the samples and the added 2000 ng/mL 142 bp dsRNA. The expected dsRNA concentration in the 2200 µg/mL sample is N/A because this sample was included as a control to account for the residual dsRNA in the mRNA material.

[mRNA Sample] (µg/mL)	[142 bp dsRNA] (ng/mL)	Expected [dsRNA] (ng/mL)	Measured [dsRNA] (ng/mL)	% Expected
2000	2000	2809	3065	109%
1500	2000	2607	3244	124%
1000	2000	2405	2863	119%
750	2000	2303	2788	121%
500	2000	2202	2723	124%
2200	N/A	N/A	890	-

## Data Availability

The raw data supporting the conclusion of this article will be made available by the authors, without undue reservation upon approval by the Merck legal department.

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
