# Peer review of "Development and Application of Automated Sandwich ELISA for Quantitating Residual dsRNA in mRNA Vaccines"

_vaccines, 2024, doi:10.3390/vaccines12080899_

Round 1

Reviewer 1 Report

Comments and Suggestions for Authors

Can reverse transcriptase qPCR be used for the quantification of dsRNA…or potentially sequencing (NGS)?  If not, could you briefly explain why not in the Introduction?

Are there USP guidelines on dsRNA testing for mRNA vaccines?  What is the lower limit of size that should be detectable by an assay, and what dsRNA LOQ should an assay have for safe manufacture of mRNA (or limit of dsRNA per drug dose)?

Was dsRNA incubation time explored as a parameter for ELISA performance optimization?

Were the K2 and J2 antibodies designed to detect a wide range of dsRNA sequences?  Do you have any general comments on what would serve as an appropriate immunogen/antigen?

What is the typical concentration of an mRNA vaccine (is it in the neighborhood of 1 mg/mL)?  Will this assay work to detect dsRNA in high concentrations of mRNA (≥ 4 mg/mL, lines 319/320)?  If not, should higher concentrations be tested in the future to demonstrate the practical application of this ELISA?

Line 94: Write out chemical name of poly I:C and why it is a potential positive control.

Line 126: consider defining 4-MUP

Line 143: Period needed at the end of this sentence.

Line 188: Suggesting changing an to a.

Lines 294-295: Would it be possible to clarify that the measured amount of residual dsRNA in the 1 mg/mL mRNA is 390 ng/mL, as shown by the unspiked result?

Line 389: Consider using a different word other than “resolve.”  Separate?

Lines 540-544: Based on these statements, do the authors feel that greater characterization work of the J2 and K2 antibodies is needed before they could be used for the purpose of monitoring dsRNA in an mRNA production process?

Author Response

Thank you very much for taking the time to review this manuscript. Your comments and suggestions were very helpful. Please find the detailed responses to your comments below along with corresponding revisions and/or corrections. In addition to calling out corrections made below, corrections were made with change tracker and highlighted in yellow.

Comment 1: Can reverse transcriptase qPCR be used for the quantification of dsRNA…or potentially sequencing (NGS)?  If not, could you briefly explain why not in the Introduction?

Response 1: Neither of these methods would be appropriate for measuring dsRNA. RT-qPCR requires specific templates that may not be present on all dsRNA species because dsRNA is a heterogeneous mixture of different sequences and lengths. NGS is low throughput and requires complex data for quantitation. More importantly, NGS would not be able to differentiate between duplex RNA and single-stranded RNA.

To address this question, we have added the following sentence to the paper (lines 71-73): “The heterogeneity of dsRNA and presence of mRNA in excess of orders of magnitude precludes some methods of measuring nucleotides such as qPCR and next generation sequencing.”

Comment 2: Was dsRNA incubation time explored as a parameter for ELISA performance optimization?

Response 2: Thank you for your input; the incubation times could be another parameter to explore to further optimize the ELISA performance. These parameters were not explored in this study because we were happy with the results we had achieved.

Comment 3: Are there USP guidelines on dsRNA testing for mRNA vaccines?  What is the lower limit of size that should be detectable by an assay, and what dsRNA LOQ should an assay have for safe manufacture of mRNA (or limit of dsRNA per drug dose)?

Response 3: The most recent USP guidelines says only that “If dsRNA is present in the mRNA vaccine, it has the potential of being immunogenic. For that reason, dsRNA content should be determined and controlled” (vaccine-mrna-guidelines-2.pdf (usp.org)). Immunoassays have typically been the workhorse for dsRNA quantitation with the dot-blot being the most used format. The specification is often <0.1% dsRNA (10,000 ng/mg) which is actually a very high concentration of dsRNA compared to what is produced from current manufacturing process discussed in the manuscript.  It is still an open question as to what a “safe” level of dsRNA would be. Without a clear answer, the industry approach has been to reduce it as much as possible. The smallest fragment that can be detected by an immunoassay is 40 bp. There is not a good way to measure species smaller than 40 bp and it is unknown what the potential safety impact of these would be.

Added text to line 55-57 as follows (edits in red): “To our knowledge, no regulatory pharmacopeia has provided target acceptable levels of dsRNA in mRNA vaccines. Existing guidance indicates that dsRNA impurities should be removed to the extent feasible, and any residual dsRNA should be well characterized and controlled18

            Ref: post-ecbs-who-regulatory-considerations-document-for-mrna-vaccines---final-version---29-nov-2021_tz.pdf

Comment 4: Were the K2 and J2 antibodies designed to detect a wide range of dsRNA sequences?  Do you have any general comments on what would serve as an appropriate immunogen/antigen?

Response 4: The K2 and J2 antibodies were designed by Schonborn et al.to detect a wide range of dsRNA sequences. The authors do not suggest methods for generating alternative antibodies. However, one could imagine isolating dsRNA produced during transcription of target mRNA sequences and using it as an immunogen to generate antibodies that are specific to the sequences of dsRNA produced as a byproduct of that specific mRNA sequence. This would require the generation of bespoke antibodies for every target mRNA sequence similar to what is done for host cell protein assays.

We have added the following to text to clarify these points (lines 74-76): “A 1991 publication by Schönborn et. al., described 4 antibodies (J2, J5, K1 and K2) that specifically bind dsRNA without binding single-stranded RNA, and are highly sensitive for a wide range of sequences that are 40 bp in length [22].”

Comment 5: What is the typical concentration of an mRNA vaccine (is it in the neighborhood of 1 mg/mL)?  Will this assay work to detect dsRNA in high concentrations of mRNA (≥ 4 mg/mL, lines 319/320)?  If not, should higher concentrations be tested in the future to demonstrate the practical application of this ELISA?

Response 5: The mRNA dose used in vaccine is determined on a product-by-product basis but is typically between 10-100 µg/dose. For example, Pfizer-BioNTech’s COVID-19 vaccine contains 30 µg of mRNA per dose and Moderna’s COVID-19 vaccine is 100 µg mRNA per dose. The dsRNA ELISA is not used to measure dsRNA in mRNA vaccines because the mRNA is vaccine is encapsulated in a lipid nanoparticle in the final drug product formulation which is incompatible with assays for dsRNA concentration. The concentration of dsRNA is instead tested in drug substance prior to encapsulation. The mRNA drug substance is typically prepared at 1-2 mg/mL. The range of mRNA concentrations during manufacturing (currently lines 331-332) were given to illustrate the need for normalizing the dsRNA concentration. They were not intended to give a range of assayable mRNA concentrations. The assay can detect dsRNA concentration in high mRNA samples by pre-dilution if necessary. 

We have modified the text to help avoid confusion for future readers (lines 331 – 332; edits in red). “The dsRNA concentration alone is insufficient to inform best manufacturing practices because the mRNA concentration of in-process samples can vary in mRNA concentration from 0.1 mg/mL to ≥ 4 mg/mL depending on the processing stage as well as variability in sample preparation.”

Comment 6: Line 94: Write out chemical name of poly I:C and why it is a potential positive control.

Response 6: Thank you for this comment, the full name of poly I:C is inadvertently never included in the original draft. The chemical name of poly I:C (Polyinosinic:polycytidylic acid) was added to the methods (line 101) and in the discussion (line 524-525). A sentence was added to the discussion to clarify why it was considered as a potential positive control (lines 526-528). “Poly I:C is structurally similar to double-stranded RNA and is commonly used as a synthetic analog to dsRNA in immune research.”

Comment 7: Line 126: consider defining 4-MUP.

Response 7: Thank you for this comment as well. The full name for 4-MUP was inadvertently never included in the original draft. The full name (4-methylumbelliferyl phosphate) was added to the first instance of the material and methods section (line 107).

Comment 8: Line 143: Period needed at the end of this sentence.

Response 8: Thank you! A period was added at the end of the sentence (line 152)

Comment 9: Line 188: Suggesting changing an to a.

Response 9: Thanks, changed, now on line 197.

Comment 10: Lines 294-295: Would it be possible to clarify that the measured amount of residual dsRNA in the 1 mg/mL mRNA is 390 ng/mL, as shown by the unspiked result?.

Response 10: The reviewer is correct that this point is difficult to understand in the original draft. The following line was added to help clarify (line 296-297) “The 1 mg/mL mRNA used to for the background contained 390 ng/mg residual dsRNA before the dsRNA spike.”

Comment 11: Line 389: Consider using a different word other than “resolve.”  Separate?

Response 11: This is a good suggestion. We have changed “resolved” to “separate” (line 402).

Comment 12: Lines 540-544: Based on these statements, do the authors feel that greater characterization work of the J2 and K2 antibodies is needed before they could be used for the purpose of monitoring dsRNA in an mRNA production process?

Response 12: Thanks for this question, we do not feel that greater characterization work of the J2 and K2 antibodies is need before they can be used for the purpose of monitoring dsRNA in mRNA production. For the purpose of mRNA manufacturing, the goal is to make an “in-control” product; we are only concerned that the apparent concentration of dsRNA does not change from one batch to the other.  This is especially true given that there has not been an empirically demonstrated impact of dsRNA on the safety or efficacy of mRNA vaccines. The dsRNA ELISA using J2 and K2 antibodies can effectively achieve this even if the reported value is not perfectly accurate. The purpose of pointing out the potential of biases was to recognize that, from a purely scientific perspective, further characterization of the J2 and K2 antibodies would enable a more nuanced understanding of the dsRNA species present.

To address this comment, we updated the conclusion as follows in the text with the following (lines 591-592; edits in red): The dsRNA sandwich ELISA protocol presented here is suitable for use in ensuring product consistency during manufacturing and could serve as a template to be iterated on as the mRNA community transitions to a universal assay to support the flourishing field of mRNA-based therapeutics.

Reviewer 2 Report

Comments and Suggestions for Authors

The manuscript presents methodological work. The authors report for the first time comprehensively the development of an automated anti-dsRNA sandwich ELISA for the quantification of residual dsRNA in adjuvanted mRNA vaccine samples.

Validation is described in great detail and has great application significance, which may facilitate the implementation of this method in control laboratories or manufacturers.

Comments:

1. Due to the large amount of detailed information, please consider including simplifications in the form of graphics - a block diagram of the test procedure with the most important information.

2. Shouldn't the A&B and C&D charts change order? First, we analyze unpurified samples, then purified ones.

Author Response

Thank you very much for taking the time to review this manuscript. Your comments and suggestions were very helpful. Please find the detailed responses to your comments below along with corresponding revisions and/or corrections. In addition to calling out corrections made below, corrections were made with change tracker and highlighted in yellow.

Comment 1: Due to the large amount of detailed information, please consider including simplifications in the form of graphics - a block diagram of the test procedure with the most important information.

Response 2: Thank you for this suggestion. A block diagram has been added to the supplementary information and is referenced on line 94)

Comment 2: Shouldn't the A&B and C&D charts change order? First, we analyze unpurified samples, then purified ones.

Response 2: The review makes a good point about the order of the figure 1 panels. It would feel natural to present the panels in an order that reflects their stage of purification. However, the observations in the unpurified samples prompted a follow-up investigation, which is discussed in the following paragraphs. We felt that the text flowed better by discussing the unpurified samples second so that we could transition directly into the investigation into matrix interference. We added the following clarifying text to address this comment (line 189) “We report first the purified samples as an example of a well-behaved sample.”

Reviewer 3 Report

Comments and Suggestions for Authors

The manuscript presents an original useful development for quality control of vaccines preparations. Its overall level accords to demands of the Vaccines journal. Nevertheless, some items in the manuscript need justifications / revisions:

1. The Abstract should contain specified composition of the detected complex and key quantitative analytical parameters of the proposed assay.

2. Lines 68-71. The specifications of the already proposed immunoassays for dsRNA should be reformulated for clear indication of their disadvantages/limitations. What means «antibodies described by Schonborn et al.» and what are their properties that determine the capabilities and disadvantages of existing immunoassays?

3. Please specify properties of the used K2 and J2 antibodies (lines 93-94). What is known about their specificity? What is their difference from the antibodies described by Schonborn et al.?

4. Line 101. The kind of the used Nunc microplates is hidden. Please give here their specification that can be found below (line 111).

5. Please comment possible mechanisms for interfering properties of proteinase K and SDS (lines 236-238). It cannot be a cross-reaction with antigen-binding sites of the used antibodies due to basically different chemical structures. Have cases of interfering effects for proteinase K or SDS on immunoassays of certain compounds been described previously in the literature?

6. mRNA constructs 1-7 are specified only in the Supplement. So their numbers at Fig. 4B are not helpful for readers. Please consider some integration of the constructs' specifications and data of their experimental studies.

7. Histograms at Figs. 4-6 should be accomplished by assessment of whether differences in the values presented in adjacent columns are statistical reliable or not.

8. Lines 437-444. The comparison with commercial ELOSA kits in testing productivity needs more detailed calculations of the discussed values (samples per hour, samples per day).

9. Lines 564-566. The statement future properties of regulations grounded by addressing to a document published in 2007 seems strange and should be checked and revised.

10. The Supplement file should contain title & authors & affiliations data for its clear identification as a separate document.

Author Response

Thank you very much for taking the time to review this manuscript. Your comments and suggestions were very helpful. Please find the detailed responses to your comments below along with corresponding revisions and/or corrections. In addition to calling out corrections made below, corrections were made with change tracker and highlighted in yellow.

Comment 1. The Abstract should contain specified composition of the detected complex and key quantitative analytical parameters of the proposed assay.

Response 1: Thank you for your comment. A sentence was added to the abstract to accommodate this requestion (lines 21-23)

The final automated sandwich ELISA was able to measure <10 ng/mL of dsRNA greater than 40 bp in length with a specificity for dsRNA over 2000-fold higher than mRNA, variability < 15% and a throughput of 72 samples per day.”

Comment 2. Lines 68-71. The specifications of the already proposed immunoassays for dsRNA should be reformulated for clear indication of their disadvantages/limitations. What means «antibodies described by Schonborn et al.» and what are their properties that determine the capabilities and disadvantages of existing immunoassays?

Response 2: Thank you for pointing out that it is not clear what is meant by "antibodies described by Schonborn et.al.” and why the sandwich ELISA is preferred over other immunoassays. The publication by Schonborn et.al. introduced 4 antibodies (J2, J5, K1 and K5) that bind dsRNA with high specificity. The antibodies are very sensitive for dsRNA of a range of sequences and lengths provided that the sequence is at least 40 bp in length. To our knowledge, these antibodies have been used in all immunoassays for dsRNA. The text from lines 68-92 has been reworded to clarify these points as follows (lines 74-80):

A 1991 publication by Schönborn et. al., described 4 antibodies (J2, J5, K1 and K2) that specifically bind dsRNA without binding single-stranded RNA, and are highly sensitive for sequences that are 40 bp in length [22]. These antibodies have been used in several quantitative immunochemical assays such as dot blot assay,[20] sandwich ELISA [16, 23] and lateral flow immunoassay [24], The sandwich ELISAs offer an appealing strategy for dsRNA quantitation because they offer an excellent balance of sensitivity, throughput, and simplicity, and can be implemented using standard equipment found in most labs.

Comment 3. Please specify properties of the used K2 and J2 antibodies (lines 93-94). What is known about their specificity? What is their difference from the antibodies described by Schonborn et al.?

Response 3: The K2 and J2 antibodies are two of the 4 antibodies described by Schonborn et.al. We thank the reviewer for pointing out that this is not clear in the original draft. We have added text to clarify this point in lines 86-87.

(edits in red) “The objective of this manuscript is to provide a template for implementing a dsRNA sandwich ELISA to support mRNA vaccine manufacturing process development using commercially available J2 and K2 antibodies along with other readily available materials and reagents.”

Comment 4. Line 101. The kind of the used Nunc microplates is hidden. Please give here their specification that can be found below (line 111).

Response 4: You are correct, this was a great catch. We have added the relevant information (line 108) “Black 96-well MaxiSorp® ELISA plates were purchased from Nunc”.

Comment 5. Please comment possible mechanisms for interfering properties of proteinase K and SDS (lines 236-238). It cannot be a cross-reaction with antigen-binding sites of the used antibodies due to basically different chemical structures. Have cases of interfering effects for proteinase K or SDS on immunoassays of certain compounds been described previously in the literature?

Response 5: Thank you for this suggestion, it was an oversight for us to not include a proposed mechanism for interference in the original manuscript. We have added the following sentence (line 246-247). ” Proteinase K and SDS can interfere with the ELISA by degrading the coating antibody by proteolysis or denaturation, respectively.”. We are not aware any other paper that has reported interference by proteinase K or SDS, but we feel that their potential to interfere is self-evident from their chemical properties.

Comment 6. mRNA constructs 1-7 are specified only in the Supplement. So their numbers at Fig. 4B are not helpful for readers. Please consider some integration of the constructs' specifications and data of their experimental studies.

Response 6: We recognize the reviewer’s position that insufficient information is shared about the mRNA constructs listed in Figure 4B. Unfortunately, we are not able to share more information about these constructs for proprietary reasons. The constructs are arbitrarily labeled 1-7 to blind their identity. The purpose of Figure 4B is to compare two manufacturing processes, the comparison was made with 7 mRNA vaccine constructs. To clarify this point, we have removed the mention of Supp. Table S1 from this section (i.e. removed “of various sequences and lengths ranging from ~2 kb in length to ~4 kb in length (Supp. Table 1)” from lines 347-348.

 Comment 7. Histograms at Figs. 4-6 should be accomplished by assessment of whether differences in the values presented in adjacent columns are statistical reliable or not.

Response 7: We appreciate the reviewer’s comment that an application of statistics would demonstrate the significance of differences or similarities in data presented in figures 4-6. However, demonstration of statistical significance is not necessary for the intentions of bar graphs presented in this paper. Figure 3 presents the data used to optimize the ELISA reagents. A robust ELISA will not have significant changes in the parameters upon changes of the reagents. Although we selected the conditions that gave the best apparent values, the fact that there is not significant changes for the other conditions is a benefit to the assay. Figure 4&5 give an example of the application of the ELISA to inform decisions during process development. We feel that for the purposes of this study a visual assessment of the differences based on the average and 1 standard deviation is sufficient to demonstrate differences. The bar graphs presented in figure 6 panel C&D are only intended to serve as an alternative way to visualize the fits of the curves presented in panels A&B respectively. The point of the figure can be impressed by visual inspection of the curves without statistical justification. No statistical evaluation has been performed on the data presented in this paper. 

 Comment 8. Lines 437-444. The comparison with commercial ELOSA kits in testing productivity needs more detailed calculations of the discussed values (samples per hour, samples per day).

Response 8: Thank you for this comment. We have modified the text in the discussion to better clarify the point we intended to make (lines 456-460). “Supporting 72 samples per day with a full titration curve using a typical 5-plate commercial kit would require opening and combining 3 kits which would impart additional variability and analyst burden from preparing reagents from multiple sources. Furthermore, the cost per sample of testing with a kit is over twice as much as testing with the in-house dsRNA ELISA described in this paper.”

Comment 9. Lines 564-566. The statement future properties of regulations grounded by addressing to a document published in 2007 seems strange and should be checked and revised.

Response 9: This is a very good catch. we had intended to reference a different WHO document (Evaluation of the quality, safety and efficacy of messenger RNA vaccines for the prevention of infectious diseases: regulatory considerations) and made a mistake when searching in EndNote. We have corrected the reference with the 2021 WHO document. (line 584)

Comment 10. The Supplement file should contain title & authors & affiliations data for its clear identification as a separate document..

Response 10: Thank you for this information. The article title, authors and affiliations have been added to the supplemental file.

Round 2

Reviewer 3 Report

Comments and Suggestions for Authors

The manuscript has been successfully revised and now may be published